# Prevalence of undernutrition and its associated factors among older adults using Mini Nutritional Assessment tool in Womberma district, West Gojjam Zone, Amhara Region, North West Ethiopia, 2020

**Amsalu Dereje Tadesse[1], Tsige Gebre Anto[2]\*, Molla Yigzaw Birhanu[2], Eskeziaw Agedew[1], Belete Yimer[1], Ayenew Negesse Abejie[1]**

1 Nutrition Department, Health Science College, Debre Markos University, Debre Markos, Ethiopia, 2 Public Health Department, Health Science College, Debre Markos University, Debre Markos, Ethiopia

\* gebretsige2007@gmail.com

## Abstract

### Background

Undernutrition is a frequent and serious problem in the world's older adults. Even though life expectancy is increasing, they are more vulnerable and at risk for nutritional problems. However, not much is known about the nutritional status of this group of the population, and they are often neglected.

### Objective

This study was aimed at assessing undernutrition and associated factors among older adults in Womberma District, West Gojjam Zone, Amhara Region, Ethiopia, 2020.

### Methods

A community-based cross-sectional study design was used among randomly selected 594 older adults aged above 60 years with a multistage simple random sampling method and proportional sample size allocation was used to address study subjects at the village level. The collected data was entered into Epi-Data version 4.2 and analyzed by using SPSS version 25. All variables with a p-value<0.25 in the bivariable analysis were considered for multivariable logistic regression for further analysis and the level of statistical significance was declared at p-value<0. 05.

### Results

The prevalence of undernutrition among older adults was found to be 14.6% (95%CI: 11.9–17.7). A number of independent variables have a significant association with undernutrition, including gender, females [(AOR (95%CI): 3.14 (1.50–6.54)], age (Oldest Old [AOR (95% CI): 4.91 (2.44–6.08)] and Middle Old, [AOR (95%CI): 2.96 (1.44–6.08)], meal frequency

**Data Availability Statement:** All the data are available within the paper and its Supporting Information files.

**Funding:** The authors received no specific funding for this work.

**Competing interests:** The authors have declared that no competing interests exist.

**Abbreviations:** AOR, Adjusted odds ratio; BMI, Body mass index; CI, Confidence interval; COR, Crude odds ratio; DDS, Dietary diversity score; GDS, Geriatrics depression scale; MNA, Mini nutritional assessment; MUAC, Mid upper arm circumference; VIF, Variance inflation factor.

[AOR (95%CI): 2.01 (1.12 (1.04–3.63)], dietary diversity score [AOR (95%CI): 2.92 (1.54–5.53)], depression [AOR (95%CI): 5.22 (3–9.07)], individuals with a sickness in the last 4 weeks [AOR (95%CI): 2.12 (1.02–4.41)] and individuals with a known hemorrhoid [AOR (95%CI): 3.51 (1.12–10.97)].

## Conclusion

This study found that the prevalence of undernutrition in older adults is high and needs attention. Sex, age, meal frequency, dietary diversity, being sick in the last 4 weeks, having hemorrhoids, and depression were the associated risk factors. Therefore, the government, family members, and other stakeholders should give more attention to older individuals.

## Introduction

Aging is a period when irreversible physiological, chronological, spiritual, and social changes and losses of roles are experienced and the adaptation of the system to the environment decreases [1]. The World Health Organization considers older adults in developing countries to be any person older than 60 years, and in developed countries, people aged over 65 years [2]. Older adults are expected to become the largest demographic group in many countries in the next few decades. The increasing number of this group gives an insight into the global community to reconsider the suitability of health infrastructure. Eleven percent of the world population (11%) and 5.0% of the Ethiopian population were categorized as older adults when aged 60 years or older [3–5].

Malnutrition in this age group is defined as "the cellular imbalance between the supply of nutrients and energy and the body's demand for them to ensure growth, maintenance, and specific functions". It may be caused by the lack of one or more nutrients (undernutrition), or an excess of nutrients (overnutrition) [6].

Currently, there is a lack of evidence and guidance on the most appropriate method of measuring the nutritional status of older adults. Even though there is no gold standard for evaluating the nutritional status of in this age group, the Mini Nutritional Assessment tool has been increasingly used worldwide for the estimation of the nutritional status [7]. The MNA tool is a short, non-invasive, reliable, and extensively evaluated nutritional assessment tool for free-living and clinically relevant older adults. It was developed by Nestlé and leading international geriatricians and is well-validated in international studies in a variety of settings. Similarly, the MNA tool is suggested by the European Society for Clinical Nutrition and Metabolism for routine geriatric nutritional assessments. The full MNA is the original version of the MNA and takes 10–15 minutes to complete. In Ethiopia, the study was conducted in Yeka sub-city, Addis Ababa, and southern Ethiopia to determine the validity and reliability of the MNA tool. The original full MNA tool was reliable and valid to identify malnourished and at risk of malnutrition among the older adults [8, 9].

Older adults are highly neglected by different health, nutrition, and social interventions carried out by different stakeholders. There is no nationally adopted guideline in Ethiopia for the management of acute malnutrition in this age group and no mandate to include them in screening for acute malnutrition. This contributes to the resistance of donors and partners, including older people as a vulnerable group. Most nutritional intervention programs are directed toward infants, young children, adolescents, and pregnant and lactating mothers. However, nutritional interventions could play a part in the prevention of degenerative

conditions in older adults and the improvement of their quality of life. Timely intervention can stop weight loss in those at risk of malnutrition. Unfortunately, not much explanation has been given for the precise estimate of undernutrition in this age group in research [10–12].

Currently, the life expectancy of Ethiopian older adults is increasing, and there are limited studies that assess the nutritional status of this age group using appropriate nutritional assessment methods. Particularly in Ethiopia, data on the nutritional status of this age group is lacking [13].

Previous studies conducted in Ethiopia include only older adults living in the urban community, but understanding the nutritional status in the rural areas is also very important to identify the most affected community to design local interventions and ensure equity in the community.

Also, most of the studies in Ethiopia, identified determinant factors of the nutritional status of this age group using a single anthropometric measurement (BMI or MUAC), but this may not be reliable to determine the nutritional status. And the MNA tool was not used for evaluating nutritional status. So, assessing the nutritional status of older adults with a valid MNA tool is very useful to identify the determinant factors of undernutrition.

In addition to this, data on the Ethiopian situation regarding the nutritional status of older adults is lacking. Rather most nutritional research evidence and intervention programs are directed toward infants, young children, pregnant and lactating mothers. Therefore, this study sets out to address the issues related to nutritional status and associated factors among older adults in the study area, and the findings of this study will have significant implications for nutrition strategy in old peoples.

## Methods

### Study design, area, and period

A community-based cross-sectional study was conducted among older adults. The study was carried out in Womberma district and its capital city is Shindy, which is located 427kms from Addis Ababa, the capital city of Ethiopia, and 172kms from Bahir Dar, the capital city of Amhara Regional State. This study was conducted between October 10 and November 1, 2020.

### Population

The source population was all older adults living in the Womberma district, and all older adults in randomly selected villages in the district were the study population. All older adults living for 6 months or longer were included in the study, whereas edematous people and those who couldn't speak properly were excluded from the study.

### Sample size and sampling procedure

The sample size was calculated using a single population proportion formula considering the margin of error (5%) and prevalence of undernutrition (22.7%) [14] among older adults (aged 60 years and over) in Debre Markos town, Amhara, Ethiopia, 2015. Taking design effect 2 and considering a 10% non-response rate, the total sample size required for this study was 594.

A two-stage stratified sampling technique was used to address the study subjects. The villages are classified as 2 urban and 19 rural. A simple random sampling technique was used to select a total of 9 villages (1 from urban and 8 from rural). Then the total sample size was proportionally allocated to each village depending on the number of older adults available in each village. The total number of older adults was identified by reviewing community health information systems/CHIS/records from the health post. First, all individuals were framed by using

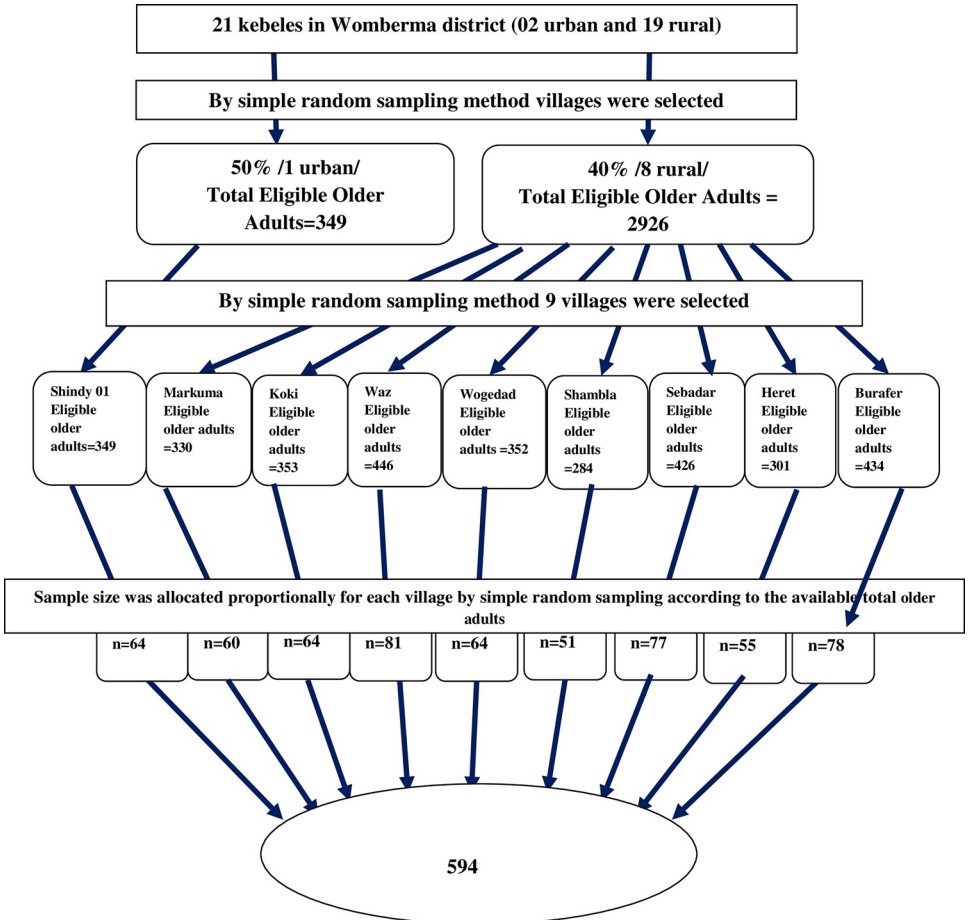

**Fig 1. Schematic representation of the sampling procedure, the prevalence of undernutrition and associated factors among older adults in Womberma district, West Gojjam, North West Ethiopia, 2020, (n = 594).**

their family folder numbers. From this list, sampled individuals were drawn through a simple random sampling technique/Computer-generated random number/, then the sampled individuals' names and particular villages were drawn from the already registered folder (Fig 1).

## Study variables

The prevalence of undernutrition is a dependent variable. The explanatory (independent) variables are age, sex, religion, marital status, family size, living arrangement, residence, occupation, educational level, wealth index, member of health insurance, common chronic medical illness (hypertension, joint pain/arthritis, asthma, diabetes mellitus), illness in the past 4 weeks, joint pain, known hemorrhoids, depression, and disability/impairments (physical impairments, vision problems, and hearing problems), dietary diversity score, time of meal (regular and irregular), meal frequency, food-cooking style/care (self, spouse, children, and/or maid), and dietary counseling/education.

## Operational definitions

**Undernutrition.** Is referring to older adults categorized as undernutrition when the total Mini Nutritional Assessment score (MNA) score is<17 points [15].

**Older adults.** Those individuals aged 60 years old. Young old: individuals' age group from 60–74 years old; Aged: those individuals from 75–84 years old; Oldest-old: individuals' age group of 85+ years old [16].

**Wealth index.** It was constructed based on variables included in household assets, size of agricultural land, and quantity of livestock and household conditions by using principal component analysis and it was classified as poor, medium, and rich.

**Household.** It refers to a group of individuals living together, typically sharing meals, shelter, resources, or a food budget, having common cooking and eating arrangements, and who are under the control of one domestic head.

**Older adults' household.** It is a household in which the older adults live for at least 6 months or above.

**Individual dietary diversity score.** It refers to food groups consumed from the total 16 food groups during the 24 hours (during day and night) before the survey. It was classified as; "Poor" (when the DDS is<3), "Medium" (when the DDS is between 4–5), and "High" (when the DDS is>6) [17].

**Edematous old age.** It is currently observable swelling from fluid accumulation in parts of the body.

**Depression.** Individuals are depressed if they score five and above and not depressed if they score below five on 15 items of the Geriatrics Depression Scale. In GDS 15 items, five questions (1, 3, 7, 11, and 13) are negative. So, if participants respond yes, they are labeled as "0" and if no, they are labeled as "1". On the other hand, positive GDS item questions were labeled as if yes, label "1" and if no, label "0" [18].

**Physical problem.** Is defined as limitation of daily life activities and restriction of participation.

## Data collection tool and procedure

**Data collection tool.** The MNA tool was administered for older adults and it was adapted from the Nestle Nutrition Institute. After extensive revision, the final version of the English questionnaire was developed and translated into Amharic.

The dietary diversity scores (DDSs) of participants were measured using the 24-hour recall method. This is the most popular method because it is less prone to recall bias and is less cumbersome for respondents. Participants were asked to list what food groups they had consumed in the past 24 hours of the survey. Sixteen food groups were used to compute the DDS of study subjects. It comprises cereals, roots and tubers, legumes and nuts, dairy products, fresh foods (meat, fish, poultry, and organ meats), eggs, vitamin A-rich fruits and vegetables, other fruits and vegetables, oils and fats, sweets, and condiments. Finally, the score was classified as; "poor" (when the DDS is<3), "medium" (when the DDS is between 4–5), and "high" (when the DDS is >6) [17].

**Personnel.** The team was comprised of one supervisor, four data collectors, and one principal investigator. Data collectors and supervisors were selected carefully and they had good knowledge of the local language, and two of them had experience in data collection.

**Procedure.** The data was collected at the community level using a questionnaire by interviewing older adults who live in the study area. Before the data collection, the respondents were informed about the purpose of the study. The data collectors give the consent paper to the respondents who can read and write, and then they mark the response in the appropriate box with their signature. For those who cannot read and write, the data collectors read the consent and provide them with a place to put their signatures. After written consent was obtained

 

and documented, the data collection was carried out. To ensure confidentiality, personal identifiers such as names and house numbers were not used during data collection.

**Anthropometric measurement.** *Weight measurement*. The study participants were weighted using a using a digital scale SECA model 874(SECA GmbH & Co. KG, Hamburg, Germany). The scale was carefully calibrated for accuracy with the use of a known weight before each measuring session and set at zero on a flat surface. The older adults were asked to stand on the scales upright and with minimum clothing and no shoes. The weight measurements were read where the needle stopped wobbling. This procedure of reading and recording the weight was repeated twice and the average weight was recorded.

*Height measurement*. The study participant stood with his or her back against the measuring board, heels, buttocks, shoulders, and head touching a flat, upright sliding headpiece. The participants' legs were placed together, making the knees and ankles touch each other. The height was recorded to the nearest 0.1 cm. For those who could not stand, the half arm-span distance (from the midline at the sternal notch to the tip of the middle finger) was taken in centimeters and doubled. The edge of the right collar bone would be located and marked. The non-dominant arm was placed in a horizontal position and in line with the shoulders. A tape measure was used to measure the distance from the midline at the sternal notch to the tip of the middle finger. The arm was confirmed to be flat and the wrists in a straight position, then readings were taken.

*The mid upper arm circumference*. The older adults were asked to bend their non-dominant arms at the elbow at a right angle with the palm up. The distance between the acromial surface of the scapula and the olecranon process of the elbow on the back of the arm was measured. The mid-point between the two was marked with a pen. They asked to let their arms hang loosely by their sides. Then the tape measure is positioned at the mid-point of the upper arm. Pinching was avoided and the measurements were recorded to the nearest 0.1 centimeters.

*The calf circumference*. The calf circumference measurements were taken from the older adults with the tape between the ankle and knee in a sitting position with the left leg. They asked to roll up their trousers to uncover the calf. A measuring tape was wrapped around the calf at the widest part and the measurement was noted to the nearest 0.1cm. The calf circumference measurements were taken with their knees bent at a 90˚ angle.

*Body Mass Index (BMI)*. Is calculated as body weight in kilograms divided by the square of height in meters.

## Data processing and analysis

One day of training was given to data collectors and supervisors. The weight scale was validated by using standardized weight before the actual weighing of each study participant. Moreover, the quality of the data was assured through careful translation and pretesting of the questionnaires. During the data collection at the end of each day, the team supervisors reviewed all questionnaires for completeness and inconsistency and ensured that missing information was added while still in the study area.

The collected data was entered into EPI DATA version 4.2 and exported to SPSS version 25 for cleaning and analysis. The outcome variable undernutrition was categorized based on the overall sum score of each subject by using the full MNA score (out of 30) and calculated using the compute command in SPSS. Descriptive statistics were employed and the findings were presented in the form of tables, charts, and graphs. The associations between dependent and independent variables were assessed and their strength was presented using an adjusted odds ratio at 95% confidence intervals. Binary logistic regression was used to assess the association between outcome and explanatory variables. Variables with a p-value<0.25 in a bivariable

 

analysis were fitted into the multivariable analysis to show the independent relationship between dependent and independent variables. Both the crude odds ratio (COR) and the adjusted odds ratio (AOR) with the corresponding 95% confidence interval (CI) were calculated to show the strength of the association. A p-value of 0.05 was used to determine if the association was statistically significant. The association between each categorical covariate and the outcome variable was tested by using a chi-square test. A principal Component Analysis was done to construct the household wealth index by asking all assets they have. The Variance inflation factor (VIF) and tolerance test were used to check the existence of outliers and multi collinearity among independent variables. The goodness of fit of the model was checked by using the Hosmer and Lemeshow test with a p-value of 0.734 [19].

**Ethical consideration.** The study was conducted after obtaining ethical clearance from Debre Markos University, College of Health Sciences, Research Ethics Committee. A formal permission letter was obtained from the Womberma District Health Office. Written consent was also obtained from the study subjects prior to data collection. The confidentiality of the information was respected throughout the data collection process. Information was given to participants about the voluntary basis of participation. Furthermore, to ensure participants' confidentiality, names or personal identifiers were not included in the written questionnaires.

## Results

### Socio-demographic characteristics

A total of 594 respondents participated in the study, with zero non response rates (Fig 2). The mean age was 68.48± 8.3 (SD) years and 53% of the study participants were females. Regarding religion, a total of 97% and 2% were orthodox Christian and Muslim followers, respectively. The majority of respondents (70%) were married, and 13% were widowed. A higher proportion of respondents (79%) cannot read and write. The majority of respondents (89.2%) were from rural areas. Most of the respondents, 83%, are currently members of health insurance. Also, regarding occupation status, 91.8% were farmers. The wealth index result showed that 33% of respondents were poor (Table 1).

**Anthropometric measurements.** Five hundred ninety-four (594) study participants' anthropometric measurements were taken. According to this, the mean weight was 53.55 ± 8.26(SD) Kgs, height was1.63 ±0.084 (SD) m, MUAC was 22.84±2.34 (SD) cm, calf circumference was 30.03 ± 2.64(SD) cm, and BMI was 20.12 ± 2.48(SD) Kg/m$^2$.

### Nutrition related characteristics of the study participant

A total of 594 respondents were screened for their dietary diversity scores. 120 (20.2%), and 153 (25.8%) had low and moderate dietary diversity, respectively. While more than half of 321 (54%) had higher dietary diversity according to the dietary diversity screening tool developed by FAO. The meal frequency was less than 3 times per day for 281 (47.3%) of the respondents, and the meal time was irregular for 227 (38.2%) of the study participants. 474 (79.8%) of the older adults were not getting dietary counseling and/or education (Table 2).

### Health-related characteristics of the study participant

Regarding the health conditions of study participants, 60 (10.1%) had been sick in the last four weeks. 146 (24.6%) had a history of chronic disease. The most common chronic medical problem was hypertension (8.2%), followed by arthritis (joint pain) (6.9%), asthma (4.7%), visual problems (4.4%), physical problems (2.5%), and known hemorrhoids (3.2%). A total of 594 respondents were screened for their depression status. 161 (27.1%) of respondents were

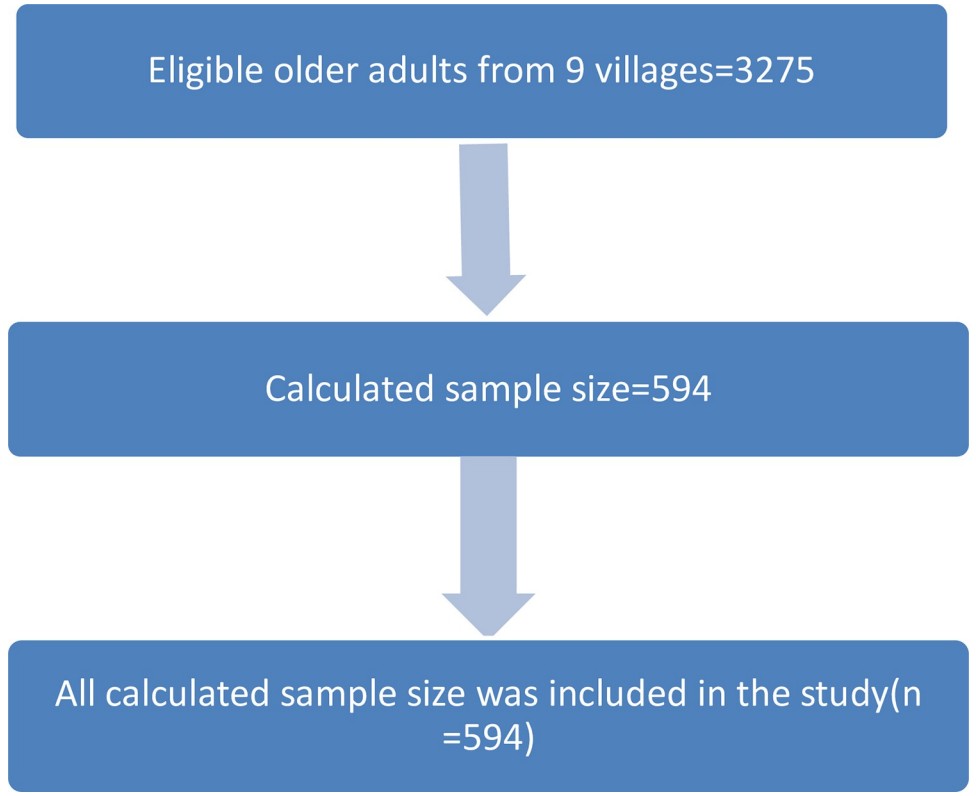

**Fig 2. Study flow diagram of prevalence of undernutrition and its associated factors among older adults using Mini Nutritional Assessment tools in Womberma district, West Gojjam Zone, Amhara Region, North West Ethiopia, 2020.**

positive for geriatric depression, while more than half, 433 (72.9%) had no depression for the geriatric depression screening tool (Fig 3).

## Prevalence of undernutrition among the older adults

The prevalence of undernutrition among older adults (aged above 60 years) was found to be 14.6% (95%CI 11.9–17.7) based on validated MNA screening tools.

## Factors associated with undernutrition

In binary logistic regression, factors such as age group, sex, not being married (widowed, divorced, and single), living alone, being rural, not being a member of health insurance, meal frequency less than three times per day, older adults not getting dietary counseling/education, self-food cooking style, irregular time of a meal, dietary diversity score, self-reported known hemorrhoids, those with visual problems, depression, and sickness in the last four weeks were found to be associated with undernutrition.

In the multivariable logistic regression, age group (middle and oldest-old age), sex (being female), meal frequency less than three times per day, low dietary diversity score, self-reported known hemorrhoids, depression, and older adults who had been sick in the last four weeks all showed significant associated with undernutrition.

Being female was nearly three times more likely to be undernourished as compared to being male [AOR (95% CI): 3.14 (1.50–6.54)]. Age was also found to be associated with

**Table 1. Distribution of socio-demographic characteristics of undernutrition among the older adults in Womberma district, North West Ethiopia, 2020 (n = 594).**

| Variables | | Total | | Undernutrition | | | |
|---|---|---|---|---|---|---|---|
| | | | | Yes | | No | |
| | | Count | % | N | % | N | % |
| Sex | Female | 315 | 53.0% | 59 | 18.7% | 256 | 81.3% |
| | Male | 279 | 47.0% | 28 | 10.0% | 251 | 90.0% |
| Age | Oldest old | 61 | 10.3% | 20 | 32.8% | 41 | 67.2% |
| | Aged | 75 | 12.6% | 18 | 24.0% | 57 | 76.0% |
| | Young old | 458 | 77.1% | 49 | 10.7% | 409 | 89.3% |
| Religion | Orthodox | 578 | 97.3% | 86 | 14.9% | 492 | 85.1% |
| | Muslim | 12 | 2.0% | 0 | 0.0% | 12 | 100.0% |
| | Protestant | 4 | 0.7% | 1 | 25.0% | 3 | 75.0% |
| Marital Status | Widowed | 77 | 13.0% | 20 | 26.0% | 57 | 74.0% |
| | Divorced | 58 | 9.8% | 13 | 22.4% | 45 | 77.6% |
| | Single | 43 | 7.2% | 8 | 18.6% | 35 | 81.4% |
| | Married | 416 | 70.0% | 46 | 11.1% | 370 | 88.9% |
| Educational Status | Unable to read and write | 469 | 79.0% | 74 | 15.8% | 395 | 84.2% |
| | Able to Read &Write | 79 | 13.3% | 8 | 10.1% | 71 | 89.9% |
| | Primary school and above | 46 | 7.7% | 5 | 10.9% | 41 | 89.1% |
| Partner Education Status | Unable to read and write | 383 | 84.7% | 40 | 10.4% | 343 | 89.6% |
| | Able to Read &Write | 47 | 10.4% | 3 | 6.4% | 44 | 93.6% |
| | Primary school and above | 22 | 4.9% | 2 | 9.1% | 20 | 90.9% |
| Occupation | No job | 14 | 2.4% | 3 | 21.4% | 11 | 78.6% |
| | Farmer | 545 | 91.8% | 78 | 14.3% | 467 | 85.7% |
| | Employee /Gov't and Private/ | 18 | 3.0% | 3 | 16.7% | 15 | 83.3% |
| | Merchant | 17 | 2.9% | 3 | 17.6% | 14 | 82.4% |
| Partner Occupation | Farmer | 412 | 91.2% | 43 | 10.4% | 369 | 89.6% |
| | Employee /Gov't and Private/ | 19 | 4.2% | 1 | 5.3% | 18 | 94.7% |
| | Merchant | 21 | 4.6% | 1 | 4.8% | 20 | 95.2% |
| Live With | Alone | 66 | 11.1% | 14 | 21.2% | 52 | 78.8% |
| | With others | 528 | 88.9% | 73 | 13.8% | 455 | 86.2% |
| Family Size | $\geq$4 | 334 | 56.2% | 49 | 14.7% | 285 | 85.3% |
| | <4 | 260 | 43.8% | 38 | 14.6% | 222 | 85.4% |
| Residence | Rural | 530 | 89.2% | 81 | 15.3% | 449 | 84.7% |
| | Urban | 64 | 10.8% | 6 | 9.4% | 58 | 90.6% |
| Health Insurance | No | 101 | 17.0% | 19 | 18.8% | 82 | 81.2% |
| | Yes | 493 | 83.0% | 68 | 13.8% | 425 | 86.2% |
| Wealth Index | Poor | 197 | 33.2% | 27 | 13.7% | 170 | 86.3% |
| | Medium | 199 | 33.5% | 29 | 14.6% | 170 | 85.4% |
| | Rich | 198 | 33.3% | 31 | 15.7% | 167 | 84.3% |

undernutrition in older adults. Being 85 years or older was nearly 5 times [AOR (95% CI): 4.91 (2.44–6.08)], while being 75–84 year age group was 3 times [AOR (95% CI): 2.96 (1.44–6.08)] more likely to be exposed to undernutrition. Those who were eating less than three meals per day were 2 times [AOR (95% CI): 2.01 (1.12–3.63)] more likely to be undernourished than those who were eating three or more meals per day.

With regards to DDS, elderly people with a low dietary diversity score were nearly 3 times more likely to be undernourished [AOR (95% CI): 2.92 (1.54–5.53)]. Depressed older adults were 5.2 times [AOR (95% CI): 5.22 (3.01–9.07)] more likely to be undernourished than those

**Table 2. Distribution of nutrition related characteristics of undernutrition among the older adults in Womberma district, North West Ethiopia, 2020 (n = 594).**

| Variables | | Total | | Undernutrition | | | |
|---|---|---|---|---|---|---|---|
| | | | | Yes | | No | |
| | | Count | % | N | % | N | % |
| Meal frequency | <3 | 281 | 47.3% | 60 | 21.4% | 221 | 78.6% |
| | ≥3 | 313 | 52.7% | 27 | 8.6% | 286 | 91.4% |
| Dietary Counseling /Education | No | 474 | 79.8% | 74 | 15.6% | 400 | 84.4% |
| | Yes | 120 | 20.2% | 13 | 10.8% | 107 | 89.2% |
| Food cooking style | Self | 221 | 37.2% | 41 | 18.6% | 180 | 81.4% |
| | Spouse | 310 | 52.2% | 39 | 12.6% | 271 | 87.4% |
| | Children | 55 | 9.3% | 6 | 10.9% | 49 | 89.1% |
| | Maid | 8 | 1.3% | 1 | 12.5% | 7 | 87.5% |
| Time of meal | Irregular | 227 | 38.2% | 41 | 18.1% | 186 | 81.9% |
| | Regular | 367 | 61.8% | 46 | 12.5% | 321 | 87.5% |
| Dietary diversity score | Low | 120 | 20.2% | 30 | 25.0% | 90 | 75.0% |
| | Middle | 153 | 25.8% | 25 | 16.3% | 128 | 83.7% |
| | High | 321 | 54.0% | 32 | 10.0% | 289 | 90.0% |

who were not depressed. Those who had been sick in the last four weeks were 2 times more likely to be undernourished [AOR (95% CI): 2.12 (1.02–4.41)]. Older adults who had a known self-reported hemorrhoid disease were nearly 3.5 times more likely to be undernourished compared to those without a known self-reported hemorrhoid disease (Table 3).

## Discussion

This study was intended to estimate the prevalence of undernutrition and identify the associated factors of undernutrition among older adults. According to this, the overall prevalence of undernutrition was found to be 14.6%. We found that female sex, age (Oldest Old and Middle Old), meal frequency, dietary diversity score, individuals who were sick in the last 4 weeks and those with a known hemorrhoid were significantly associated with undernutrition.

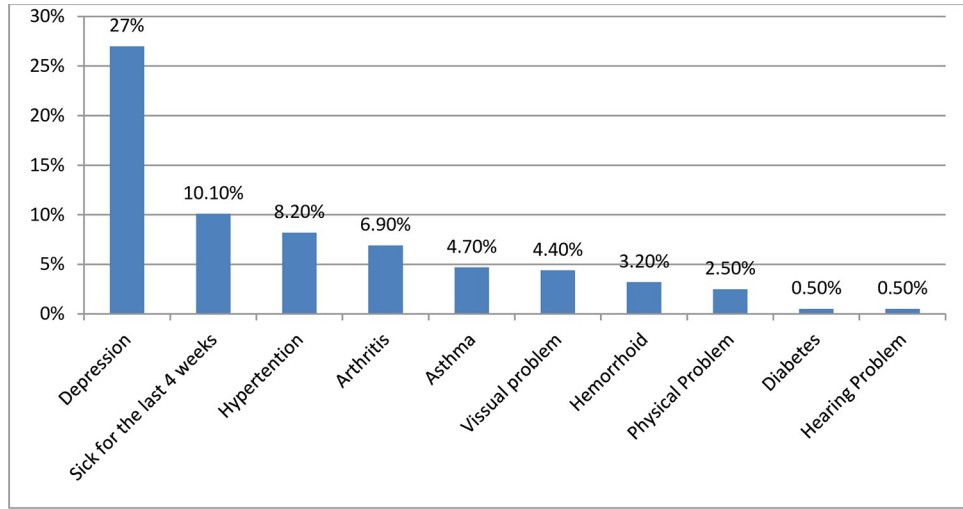

**Fig 3. Health-related characteristics with undernutrition among older adults in Womberma district, North West Ethiopia, 2020 (n-594).**

**Table 3. Bivariable and multivariable logistic regression of factors associated with undernutrition among the older adults in Womberma district, North West, Ethiopia, 2020 (n = 594).**

| Variables | | Undernutrition | | COR(95%CI) | AOR(95%CI) |
|---|---|---|---|---|---|
| | | Yes | No | | |
| | | N | N | | |
| Sex | Female | 59 | 256 | 2.07 (1.28–3.35) | 3.13 (1.50–6.54)** |
| | Male | 28 | 251 | 1 | 1 |
| Age | Oldest old | 20 | 41 | 4.07 (2.21–7.50) | 4.91 (2.44–6.08)* |
| | Aged | 18 | 57 | 2.64 (1.44–4.84) | 2.96 (1.44–6.08)** |
| | Young old | 49 | 409 | 1 | 1 |
| Marital Status | Widowed | 20 | 57 | 2.82 (1.56–5.11) | 1.34 (0.62–2.92) |
| | Divorced | 13 | 45 | 2.32 (1.17–4.63) | 0.98 (0.40–2.41) |
| | Single | 8 | 35 | 1.84 (0.80–4.20) | 2.96 (0.99–7.89) |
| | Married | 46 | 370 | 1 | 1 |
| Live With | Alone | 14 | 52 | 1.68 (0.88–3.18) | 0.52 (0.20–1.36) |
| | With others | 73 | 455 | 1 | 1 |
| Residence | Rural | 81 | 449 | 1.74 (0.73–4.18) | 1.59 (0.60–4.23) |
| | Urban | 6 | 58 | 1 | 1 |
| Health Insurance | No | 19 | 82 | 1.45 (0.83–2.54) | 1.14 (0.56–2.33) |
| | Yes | 68 | 425 | 1 | 1 |
| Meal frequency | <3 | 60 | 221 | 2.88 (1.77–4.68) | 2.01 (1.12–3.63)** |
| | ≥3 | 27 | 286 | 1 | 1 |
| Dietary Counseling /Education | No | 74 | 400 | 1.52 (0.81–2.85) | 0.88 (0.41–1.88) |
| | Yes | 13 | 107 | 1 | 1 |
| food cooking style | Self | 41 | 180 | 1.59 (0.19–13.32) | 0.83 (0.05–12.78) |
| | Spouse | 39 | 271 | 1.01 (0.12–8.41) | 1.24 (0.08–18.38) |
| | Children | 6 | 49 | 0.86 (0.09–8.22) | 0.27 (0.02–4.72) |
| | Maid | 1 | 7 | 1 | 1 |
| Time of meal | Irregular | 41 | 186 | 1.54 (0.97–2.43) | 0.83 (1.46–0.50) |
| | Regular | 46 | 321 | 1 | 1 |
| Dietary diversity score | Low | 30 | 90 | 3.01 (1.73–5.23) | 2.92 (1.54–5.53)* |
| | Middle | 25 | 128 | 1.76 (1.01–3.10) | 1.34 (0.69–2.59) |
| | High | 32 | 289 | 1 | 1 |
| Hemorrhoids | Yes | 6 | 13 | 2.82 (1.04–7.62) | 3.51 (1.12–10.97)** |
| | No | 81 | 494 | 1 | 1 |
| Visual problem | Yes | 10 | 16 | 3.99 (1.75–9.10) | 2.29 (0.85–6.18 |
| | No | 77 | 491 | 1 | 1 |
| Sick the last 4 week | Yes | 17 | 43 | 2.62 (1.42–4.85) | 2.12 (1.02–4.41)** |
| | No | 70 | 464 | 1 | 1 |
| Depression | Depression | 54 | 107 | 6.12 (3.78–9.91) | 5.22 (3.01–9.07)* |
| | No depression | 33 | 400 | 1 | 1 |

AOR = Adjusted Odds Ratio; CI = Confidence Interval, COR = Crude Odds Ratio

* = p-value<0.001

** = p-value<0.05

The prevalence of undernutrition was in line with studies done in Harari region, Eastern Parts of Ethiopia (15.7%), Sodo district, Wolaita zone, Ethiopia(17.1%), Aykel town administration (17.6%), and Sri Lanka (12.6%) [13, 20–22]. However, it was higher than the studies conducted in Bole sub-city, Addis Abeba, Ethiopia (6.2%),Niamey-Niger (7.8%), Turkey

(8.4%), and southwest China (3.2%) [23–26]. This difference might be due to variation in a geographical area, socio-economic status, and low level of education. Moreover, the absence of feeding diversified foods and seasonal variation could also contribute to such a difference.

This study revealed that undernutrition was higher among females than males. Females were nearly three times more likely to be undernourished than males. This is comparable to studies conducted in Debre Markos and Gonder, Ethiopia [11, 14]. The reason could be that older females continue tomcare for their grandchildren while receiving less care for themselves. In addition to this, repeated pregnancies, high workload, cultural beliefs, gender discrimination and low social freedom to access income-generating activities may influence women's nutritional status [27]. This is also scientifically supported that, women are more likely to be exposed to nutritional deficiencies than men, for reasons including women's reproductive biology, low social status, poverty, and lack of education [28]. But no significant relationship was noted between undernutrition and the sex of the individual in Lattakia, Syrian Arab Republic. This might be due to geographic variation and socio-economic status of women [29].

Age 85 years and above and between 75–84 years was found to be a significant factor of undernutrition. This finding was similar to a study done in Debre Markos and Gondar, Ethiopia [11, 14]. This could be because as age increases, body composition also changes, and there could be a loss of fat and muscle which results in malnutrition. This is scientifically supported that as age increases, the vulnerability to undernutrition also increases due to the natural aging process accompanied by physiological and functional changes. Acute and chronic illness with poor prognosis can impact nutritional status and cause inadequate nutrition [10, 30]. Regarding meal frequency, there was a significant association with elderly undernutrition. This result was consistent with a study done in Debre Markos town and Bangladesh [14, 31]. This change in meal pattern could occur as the result of natural aging processes and physiological or physical changes, and it may be due to low access to food. The dietary diversity score in this study was found to be positively associated with undernutrition. This was supported by studies done in Gonder and Aykel town, Ethiopia. This could be due to a lower economic status making it difficult to afford and access varied types of food [11, 21]. Even though there are surplus food items in the study area, it may be related to poor utilization and undiversified consumption habits. Food availability alone in an area cannot guarantee diverse feeding habits [32]. Another important finding in this study was depression, which showed a significant association with undernutrition among older adults. Not being involved actively in work would make them less active and at risk for undernutrition. Depression could change eating behaviors and, influences people's appetite and food intake, which could lead to an increased risk of undernutrition. This finding is supported by reports from Harergie, east Ethiopia, which illustrated a significant relationship between depression and undernutrition [20].

In this study, sickness in the last four weeks was a significant factor. This could be a change in metabolic needs (disease and increases), in addition to the patho physiologic effects of the disease that may affect the gastrointestinal tract and lead to loss of appetite, difficulties in chewing or swallowing, digestion, absorption, and metabolism. It can also have an impact on their ability to prepare food and feed themselves, as they may become bed- or chair-bound and loss motivation and desire to eat [33]. This finding is supported by reports from study in Lattakia, Syrian Arab Republic [29].

In this study, 24.6% of study participants had a history of chronic conditions including hypertension, arthritis (joint pain), asthma, visual problems, known hemorrhoids, and physical problems. But among these conditions only hemorrhoids show a significant association with undernutrition. The possible justification could be that an older individual with hemorrhoids might not feed adequately due to fear of pain during defecation and may lose appetite as a result of the pain. In contrast to this, studies conducted in Turkey, Iran, and Austria

reported that chronic conditions like hypertension, stroke, diabetes, heart disease, cancer, musculoskeletal disorders, and respiratory syndrome were significantly associated with undernutrition [34–36]. Our study has assessed the prevalence of undernutrition in the most neglected and vulnerable groups of population using a sufficient sample size. Yet, this study has the following limitations: the cross-sectional nature of the survey makes it difficult to see the temporal variation; and a single 24-hour recall of dietary data might not reflect the usual intake of participants to measure the dietary diversity score.

## Conclusion and recommendations

The overall prevalence of undernutrition in the Womberma district was high and factors including sex, age (middle and oldest-old age), meal frequency, low dietary diversity, sickness in the last 4 weeks, having a known self-reported hemorrhoid, and depression were significantly associated with undernutrition. Therefore policy makers should incorporate older adults' nutrition within the DHS survey. The government should give special attention and support to female older adults. Nutritional supplementation, care and support should be given to middle and oldest-old age groups, sickness in the last 4 weeks, hemorrhoid patients, and those with depression. Nutritional counseling programs should be strengthened to improve meal frequency and dietary diversity. The community should utilize local existing food sources and increase crop diversity to enhance dietary diversity. Family members and caregivers should closely follow the feeding habits of older adults to improve meal frequency.

## Supporting information

**S1 Data. Older adults' undernutrition data.**
(SAV)

## Acknowledgments

First of all, we would like to thank Womberma district Health Office authorities for their permission to conduct this study and our study participants for their cooperation in providing valuable information.

Secondly, our heartfelt gratitude goes to our data collectors and supervisors for giving us their valuable time.

Lastly, we would like to thank Debre Markos University for providing us with the chance to conduct this study and for its technical support.

## Author Contributions

**Formal analysis:** Amsalu Dereje Tadesse.

**Funding acquisition:** Molla Yigzaw Birhanu.

**Methodology:** Amsalu Dereje Tadesse, Eskeziaw Agedew, Belete Yimer, Ayenew Negesse Abejie.

**Software:** Eskeziaw Agedew.

**Supervision:** Eskeziaw Agedew, Belete Yimer, Ayenew Negesse Abejie.

**Validation:** Amsalu Dereje Tadesse.

**Writing – original draft:** Amsalu Dereje Tadesse.

**Writing – review & editing:** Tsige Gebre Anto, Molla Yigzaw Birhanu.

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
