## [Decision Letter · Decision Letter 0]

2 Feb 2022

PONE-D-21-38283Prevalence of undernutrition and its associated factors among elderly people using Mini Nutritional Assessment tools in Womberma District, West Gojjam Zone, Amhara Region, North West Ethiopia, 2020 .PLOS ONE

Dear Dr. Gebre,

Thank you for submitting your manuscript to PLOS ONE. After careful consideration, we feel that it has merit but does not fully meet PLOS ONE’s publication criteria as it currently stands. Therefore, we invite you to submit a revised version of the manuscript that addresses the points raised during the review process.

We look forward to receiving your revised manuscript.

Kind regards,

Subhendu Kumar Acharya, Ph.D

Academic Editor

PLOS ONE

Journal Requirements:

- https://applications.emro.who.int/emhj/v21/10/EMHJ_2015_21_10_753_761.pdf?ua=1&ua=1

- https://www.hindawi.com/journals/bmri/2020/8855276/

In your revision ensure you cite all your sources (including your own works), and quote or rephrase any duplicated text outside the methods section. Further consideration is dependent on these concerns being addressed.

Reviewers' comments:

Reviewer's Responses to Questions

**Comments to the Author**

1. Is the manuscript technically sound, and do the data support the conclusions?

Reviewer #1: Partly

Reviewer #2: Partly

2. Has the statistical analysis been performed appropriately and rigorously? 

Reviewer #1: Yes

Reviewer #2: No

3. Have the authors made all data underlying the findings in their manuscript fully available?

Reviewer #1: Yes

Reviewer #2: Yes

4. Is the manuscript presented in an intelligible fashion and written in standard English?

Reviewer #1: Yes

Reviewer #2: No

5. Review Comments to the Author

Reviewer #1: The study is well justified with a primary objective of estimating the prevalence of undernutrition/malnutrition among the older adults in Ethiopia.

While the strengths of the study include a very limited objective, in vulnerable study population population and use of validated standardized tools, the study needs further improvements before it can be published. The following suggestions may be of help to the authors:

1. While I was able to understand the artionale of the study, a more explicit description in a few lines would definitiely help. Why was this study done in this particular setting/population and why now ? What were the previous gaps in knowledge identified and how thsi study would like to fill those gaps ? Has previous evidence synthesis/reviews identified the need for such a study in this population ?

2. The authors may think of adding secondary objectives to the study (correlates of malnutrition ?)

3. I would like to see more justification on the sample size calculations. Specifically for the design effect of 2 and a non-response of 10%. As I can see, the final sample size is same as the calculated one and that implies ZERO non-response?

4. A graphical representation of the sampling frame/sampilng methods would he useful to the readers as well.'

5. The statistical analysis is to be further justified. The reasoning for the selection of variables (p value <0.25 ?), the actual multivariate model used , the predictor variable selection (possible correlates) and the goodness of fit estimate cut offs need further explanation. Did the authors find multicollinearity ? What was the finding of VIF analyses ?

6. It may be useful to start the results with a Study flow diagram- how many households were approached to how many finally included. Maybe the authors can look at some standard templates for such cross sectional surveys.

7. Did the authors analyse further the association between chronic conditions and nutritional status? Was multimorbidity associated with malnutrition ? What chronic conditions seem to be predictive of malnutrition ? This would be an interesting analyses and add value.

8. The discussion needs to focus more on the uniqueness of the findings of this study. As the authors themseleves quote local studies in similar settings with similar results, what was the added benefit of this study needs to be made more clearer. Boradly speaking, the discussion needs revisions as I felt the readers would have to be provided with a more compact and less verbose dicussion on major findings. Comparisons with all previousl studies seem unncessary at some points. Additionally a more in depth discussion on the role of chronic conditions (not just hemorrhoids) may be of interest.

9. Why was only verbal consent obtained? I feel such studies in vulnerable groups ask for written informed consent as per most national and international guidelines on health research. A justification for the same is warranted.

10. Was anemia assessed and if so a detailed discussion on its association with malnutrition may be added.

11. The recommnedations need to be based on the findings of the study in question and not gerenal observations and dissemination of knowledge. I suggest the authors to narrow down their recommendations and focus on what they found alone.

12. A section on potential strengths and limitations of the study is required in the discussion section and may be added.

13. On some minor issues, I suggest tha authors to stick to the terminology of 'older adults' throughout the manuscript and avoid the term 'elderly' which is being less frequently used in current literature to describe this population group. Further additional references on chronic condition and malnutritions may be added. The utility of figure-2 is questionable and so may be removed as it has been described in the text. The prevalence of depression seems high. Any discussion on this would be useful for a group of readers. The spelling of hypertension, mellitus, arthritis and hemorrhoids may be checked in the figure. Also 'physical problems' may need more explanation on operational definition.

I feel the study holds merit and may be of scienntific interest. However I feel the above observations need to be addressed before it can be taken up for publication.

Reviewer #2: Major comments

1. The article provides important information on nutritional problems in the elderly in North West Ethiopia. However, the conclusions by the authors are too broad and is far reaching that is beyond the objective of the study.

2. The authors may focus why existing surveys such as DHS do not undertake nutrition surveys in this vulnerable population and what are the limitations (if any) and how the DHS can include this vulnerable age groups.

3. This will provide much needed rational for the incorporation of elderly population within the DHS surveys and will be of great help for recommendations for the policy makers across the developing world.

4. The statistical analysis needs to adjust for the cluster design adopted and it is unclear whether this was taken into account for the analysis.

5. BMI is a simple indicator for assessing nutritional deficiencies in older adults but the authors do not mention the same and how it compares with the existing method.

Minor Comments

Abstract

1. In methods section, rephrase the sentence. ” kebeles” may be replaced by a generic word such as “ward or village”.

2. Rephrase the last sentence in methods section and correct the word “statistical”. Use either 95% CI or P value less than 0.05 but not both as they represent the same.

3. Rephrase the last line of conclusions and do not make it general but specific to the study.

Paper

1. In introduction, Para 4, last line may be rephrased from “It is a good way to start the day” and provide context specific information.

2. In methods, the model and make used for anthropometric measurements needed to be mentioned (e.g. SECA 874 flat weighing scale, Hamburg, Germany).

3. In methods, how many had armspan measurements were taken and how many heights were measured need to be reported.

4. In the methods section, for the last para, “The goodness of fit of the model was checked by using the Hosmer and Lemeshow test with a p-value of 0.734.” needs an appropriate reference.

5. In results section, details on weight, arm span, height, weight and BMI may be presented.

6. In discussion, limitations and strengths of the study may be discussed.

7. In conclusions, major comments may be used as the basis for write-up.

8. References need to be formatted properly and should include journal name, issue number as per the journal requirements.

6. PLOS authors have the option to publish the peer review history of their article (what does this mean?). If published, this will include your full peer review and any attached files.

Reviewer #1: **Yes: **Jaya Singh Kshatri, MD

Reviewer #2: **Yes: **Dr. Raja Sriswan Mamidi

---

## [Author Response · Author response to Decision Letter 0]

17 May 2022

Editor’s comment 

1. Your manuscript meets the PLOS ONE’S style

2. About occurrence of overlapping text 

3. About data availability

4. Changes to data availability statement

5. About ethical statement

 Answers to Editor: thank you dear editor for your constructive comments!

1. Thank you, we have prepared the manuscript according to the PLOS ONE’S style and please see the improvement of the manuscript.

2. Thank you, some overlapping texts are removed and corrected. Sources of information are also cited. Please see the improvement of the manuscript

3. Thank you,

 a. Data does not contain potentially sensitive information and shared as supporting information files. 

 b. the minimal anonymzed data set are uploaded as supporting information files

4. Thank you dear editor, we have changed the data availability statement. Please see page 18 line 416

5. Thank you, the ethical statement is written only on the method section of the paper and see page 9 line 248. 

Reviewers’ comments:

Comments to the Author

1. Is the manuscript technically sound and do the data support the conclusions?

2. Has the statistical analysis been performed appropriately and rigorously?

3. Have the authors made all data underlying the findings in their manuscript fully available?

4. Is the manuscript presented in an intelligible fashion and written in Standard English?

Answers to reviewers: thank you dear reviewers

1. Thank you, we tried to improve the manuscript and conclusion sections based on the finding of the report and scientific data & please see the improvement of the manuscript and conclusion section. 

2. Thank you dear reviewers, the statistical analysis have been conducted appropriately and rigorously, see regression table. 

3. Thank you, the summary statistics are well written on the result section. Additionally, we have shared data publicly as supporting information files.

4. Thank you, we tried to present the manuscript in intelligible fashion and Standard English. Please see the improvement of the manuscript.

Review Comments 

Reviewer#1

1. About the rational of conducting this study

2. About adding secondary objective to the study (correlates of malnutrition?)

3. Justification on the sample size calculations 

4. Graphical representation of the sampling method

5. Further justification of the statistical analysis(selection of variables (p value <0.25 ?), the actual multivariate model used, the predictor variable selection (possible correlates), the goodness of fit estimate, Multicollinearity and VIF analyses 

6. About study flow diagram 

7. About analysis of the association between chronic conditions and nutritional status

8. Revision of the discussion part 

9. Why was only verbal consent obtained?

10. Was anemia assessed and if so a detailed discussion on its association with malnutrition may be added.

11. About revision of the recommendations

12. Adding the strengths and limitations of the study in the discussion section

13. About the correction of terminology, additional references on chronic condition and malnutrition, about utility of Figure 2, discussion on the prevalence of depression, spelling error and operational definition of physical problems 

Answers to Reviewer#1 

Thank you dear reviewer

1. Thank you, the rational of conducting this study is well addressed on the manuscript. Please see page 4 line 98-111

2. Thank you, the secondary objective of this study is factors associated with undernutrition and written on study objective (on abstraction section) and result section. 

3. Thank you, initially the calculated sample size was 270 by considering p(prevalence of undernutrition among old age people from previous study)= 22.7% and d(margin of error)=5%. Then by taking into account design effect of 2 and 10% non-response rate the final sample size required of this study was 594 and see page 6 line 142-145. The non-response rate was Zero in this study because the data collectors were well trained and they visited the selected older adults’ household three times.

4. Thank you, we have showed the graphical representation of the sampling method/procedure. Please see figure 1

5. Thank you, Hosmer and Lemeshow applied statistics recommend that in Bivariable analysis variables with p value<0.25 is a candidate for multivariable analysis. In multivariable analysis variables with p value < 0.05 are declared to be significantly associated with the outcome variable. In our study, we used Hosmer and Lemeshow test of goodness of fit model. To say the model is fit p value >0.5. In our case p value is 0.734. We have checked for Multicollinearity and the existence of outliers between independent variables so there are no Multicollinearity effects between variables. The finding of VIF was 1.045 and tolerance test was 0.957 

6. Thank you, we have tried to show the study flow diagram and please see figure 2

7. Thank you, we analyzed the association between chronic conditions and malnutrition. Among respondents, 146 (24.6%) had a history of chronic diseases including hypertension (8.2 %), arthritis (joint pain) (6.9%), asthma (4.7%), visual problem (4.4%), physical problems (2.5%), diabetes (0.5%) and known hemorrhoids (3.2%). In our analysis, only hemorrhoids that significantly associated with malnutrition and please see regression table(Table 3) 

8. Thank you, we tried to edit and re write the discussion part of the manuscript based on received comments please see page 15 line 336-414

9. Thank you, this was wrongly written. We had prepared written consent. For those who can read and write, the data collectors gave them the consent to read and put their signature prior to data collection whereas for those who couldn’t read and write, the data collectors read the consent and provide them the consent to put their signature. Please see page 8 line 209-214

10. Thank you, since this is a population based study, we couldn’t assess anemia by asking the participants without any evidence (laboratory result). Additionally, we couldn’t afford the laboratory equipment because this study had no funding body.

11. Thank you, the recommendation part is revised and corrected please see page 18 line 429-436

12. Thank you, the strength and limitation of the study have been added in the discussion section please see page 18 line 415-419

13. Thank you dear reviewer, we tried to address these all issues please see the improvement of the manuscript. 

Reviewer #2

Major comments

1. Regarding the conclusions

2. Justification on why existing surveys such as DHS do not undertake nutrition surveys in this vulnerable population 

3. About recommendations for the policy makers incorporating elderly population nutrition within the DHS surveys 

4. Adjustment of the statistical analysis for the cluster design 

5. Justification for why the authors didn’t use BMI as an indicator for assessing nutritional deficiencies in older adults

Minor comments

Abstract

1. About rephrasing the word Kebeles

2. About correcting the word “Statistical”

3. Rephrasing the last line of conclusions

 Paper

1. Rephrasing the last line of paragraph 4 of the introduction section

2. Mentioning the anthropometric measurements 

3. How many had arm span measurements and heights been measured need to be reported?

4. Reference for goodness of fit model with a p-value of 0.734

5. Details presentation on weight, arm span, height and BMI 

6. Discussing limitations and strengths of the study on the discussion part

7. Incorporating major comments in conclusions

8. Formatting the references properly 

Answers to Reviewer#2

 Thank you dear reviewer,

Answer for major comments

1. Thank you, the conclusion section is re written please see page 18 line 425-429

2. Thank you, DHS in Ethiopia focus on the nutrition status of mothers, children and adults aged 15 to 49 (using BMI). But it doesn’t incorporate the nutritional status of these vulnerable groups of population.

3. Thank you, we have recommended the policy makers about incorporating the older adults’ nutrition within the DHS survey. Please see page 19 line 429

4. Thank you dear reviewer, there is no significance difference of older adults’ undernutrition across the clusters since the p value=0.6

5. Thank you, the authors used BMI as one part in MNA tool and mentioned under anthropometric measurements. Please see page 9 line 246

Answer for minor comments

Thank you dear reviewer,

 Abstract

1. Thank you, the word “kebeles” is rephrased as “Village”. Please see page 2 line 38

2. Thank you, we have corrected the word “statistical” as P value less than 0.05 and see page 2 line 42

3. Thank you , we have corrected the last line of the conclusion and see page 2 line 54

 Paper

1. Thank you dear reviewer, we have removed the sentence since it is not important.

2. Thank you, we measured weight of the participants using a digital scale SECA model 874(SECA GmbH & Co. KG, Hamburg, Germany). Please see page 8 line 217

3. Thank you, we measured the arm span and height twice per individual and reported the average.

4. Thank you, we have cited the reference and please see 10 line 274

5. Thank you, we have addressed these issues on the result section and please see page 12 line 299 

6. Thank you, the strength and limitation of the study have been added in the discussion section please see page 18 line 415-419

7. Thank you, We have incorporated major comments in the conclusion and see page 18 line 424

8. Thank you, the references are properly formatted and see the improvement of the references. 

6. Do you want your identity to be public for this peer review? 

Answer: Yes

---

## [Decision Letter · Decision Letter 1]

31 Aug 2022

Prevalence of undernutrition and its associated factors among older adults using Mini Nutritional Assessment tools in Womberma District, West Gojjam Zone, Amhara Region, North West Ethiopia, 2020 .

PONE-D-21-38283R1

Dear Dr. Anto,

We’re pleased to inform you that your manuscript has been judged scientifically suitable for publication and will be formally accepted for publication once it meets all outstanding technical requirements.

Kind regards,

Subhendu Kumar Acharya, Ph.D

Academic Editor

PLOS ONE

Additional Editor Comments (optional):

Reviewers' comments:

Reviewer's Responses to Questions

**Comments to the Author**

1. If the authors have adequately addressed your comments raised in a previous round of review and you feel that this manuscript is now acceptable for publication, you may indicate that here to bypass the “Comments to the Author” section, enter your conflict of interest statement in the “Confidential to Editor” section, and submit your "Accept" recommendation.

Reviewer #2: All comments have been addressed

Reviewer #3: All comments have been addressed

2. Is the manuscript technically sound, and do the data support the conclusions?

Reviewer #2: Yes

Reviewer #3: Yes

3. Has the statistical analysis been performed appropriately and rigorously? 

Reviewer #2: Yes

Reviewer #3: (No Response)

4. Have the authors made all data underlying the findings in their manuscript fully available?

Reviewer #2: Yes

Reviewer #3: Yes

5. Is the manuscript presented in an intelligible fashion and written in standard English?

Reviewer #2: Yes

Reviewer #3: Yes

6. Review Comments to the Author

Reviewer #2: I thank the authors for incorporating all my suggestions and addressing my comments.

Minor comment

Line 72: ofin to be separated as of in.

While, a complex survey analysis could have been performed, the conclusions of the paper are far reaching on the policy makers to include old age groups in DHS surveys and is recommended for publication. This is a seminal paper that needs to be highlighted and flagged in future meetings of DHS.

Reviewer #3: Thanks, the authors for addressing all the comments provided in first round of review. However, the manuscript needs minor English revision.

7. PLOS authors have the option to publish the peer review history of their article (what does this mean?). If published, this will include your full peer review and any attached files.

Reviewer #2: **Yes: **Raja Sriswan Mamidi

Reviewer #3: **Yes: **Krushna Chandra Sahoo, ICMR-Regional Medical Research Centre, Bhubaneswar, India

---

## [Editor Report · Acceptance letter]

15 Feb 2023

PONE-D-21-38283R1 

Prevalence of undernutrition and its associated factors among older adults Using Mini Nutritional Assessment Tool in Womberma District, West Gojjam Zone, Amhara Region, North West Ethiopia, 2020. 

Dear Dr. Anto:

I'm pleased to inform you that your manuscript has been deemed suitable for publication in PLOS ONE. Congratulations! Your manuscript is now with our production department. 

Kind regards, 

on behalf of

Dr. Subhendu Kumar Acharya 

Academic Editor

PLOS ONE